# Prognosis Communication in Pediatric Oncology: A Systematic Review

**DOI:** 10.3390/children10060972

**Published:** 2023-05-30

**Authors:** Estera Boeriu, Alexandra Borda, Eunice Miclea, Amalia-Iulia Boeriu, Dan Dumitru Vulcanescu, Iulia Cristina Bagiu, Florin George Horhat, Alexandra Flavia Kovacs, Cecilia Roberta Avram, Mircea Mihai Diaconu, Luiza Florina Vlaicu, Otniel Dorian Sirb, Smaranda Teodora Arghirescu

**Affiliations:** 1Department of Pediatrics, “Victor Babes” University of Medicine and Pharmacy, Eftimie Murgu Square 2, 300041 Timisoara, Romania; estera.boeriu@umft.ro (E.B.); sarghirescu@yahoo.com (S.T.A.); 2Department of Oncology and Haematology, “Louis Turcanu” Emergency Clinical Hospital for Children, Iosif Nemoianu Street 2, 300011 Timisoara, Romania; borda.alexandra@gmail.com (A.B.); avram.eunice@yahoo.ro (E.M.); 3Anaesthesiology and Intensive Care Department, Klinikum Rechts der Isar Der Technischen, Universitat Munchen, Ismaninger Street 22, 81675 Munchen, Germany; boeriu.amalia@gmail.com; 4Multidisciplinary Research Center on Antimicrobial Resistance (MULTI-REZ), Microbiology Department, “Victor Babes” University of Medicine and Pharmacy, Eftimie Murgu Square 2, 300041 Timisoara, Romania; dannvulcanescu@gmail.com (D.D.V.); oti93@yahoo.com (O.D.S.); 5Department of Oncology, Onco-Help Association, Ciprian Porumbescu Street 56-59, 300239 Timisoara, Romania; alexandra.kovacs@unitbv.ro; 6Department of Residential Training and Post-University Courses, “Vasile Goldis” Western University, Liviu Rebreanu Street 86, 310414 Arad, Romania; avram.cecilia@uvvg.ro; 7Department of Obstetrics and Gynecology, “Victor Babes” University of Medicine and Pharmacy, 300041 Timisoara, Romania; diaconu.mircea@umft.ro; 8Department of Social Assistance, Faculty of Sociology and Psychology, Western University, Vasile Parvan Boulevard 4, 300223 Timisoara, Romania; florina.vlaicu@e-uvt.ro; 9Eduard Pamfil Psychiatry Clinic Timisoara, Iancu Vacarescu Street 21, 300425 Timisoara, Romania

**Keywords:** prognosis communication, pediatric oncology, children with cancer, parents, caretakers, physicians

## Abstract

Background: While communication plays an important role in medicine, it also often represents a challenge when the topic at hand is the prognosis of a high-risk condition. When it comes to pediatric oncology, the challenge becomes even greater for physicians who have to adapt their discourse to both the child and their family. Methods: Following the PRISMA guidelines, an advanced search on PubMed, Scopus and the Cochrane Library was performed, from 1 January 2017 to 31 October 2022. Demographic data for caregivers, pediatric patients and physicians were extracted, as well as diagnosis, prognosis, presence at discussion, emotional states and impact on life, trust, decision roles, communication quality and other outcomes. Results: A total of 21 articles were analyzed. Most studies (17) focused on caregivers, while only seven and five studies were focused on children and physicians, respectively. Most parents reported high trust in their physicians (73.01%), taking the leading role in decision making (48%), moderate distress levels (46.68%), a strong desire for more information (78.64%), receiving high-quality information (56.71%) and communication (52.73%). Most children were not present at discussions (63.98%); however, their desire to know more was expressed in three studies. Moreover, only two studies observed children being involved in decision making. Most physicians had less than 20 years of experience (55.02%) and reported the use of both words and statistics (47.3%) as a communication method. Conclusions: Communication research is focused more on caregivers, yet children may understand more than they seem capable of and want to be included in the conversation. More studies should focus on and quantify the opinions of children and their physicians. In order to improve the quality of communication, healthcare workers should receive professional training.

## 1. Introduction

In medicine, communication is of paramount importance in establishing a rapport with patients and their families. It is also vital to communicate effectively in order to convey accurate information to patients and their families, so that they can make an informed decision about their healthcare. However, disclosing the diagnosis of a serious illness or the poor prognosis of a disease often represents a daunting task for physicians and a life-changing event for patients and their families [1,2]. In the past, before the 1960s, when clinicians began to promote a more open approach regarding communication with their patients, most literature recommended protecting patients, especially pediatric patients, in order to shield them from distress [2,3].

Now, open communication, which sheds light upon diagnosis and prognosis and includes the child’s parents in the care team, is a much more widely used approach. Recent studies have shown the benefits of discussing the diagnosis with pediatric patients themselves, using age-appropriate language [1,4,5]. This, in return, eases the burden on the parents of having to maintain a façade and opens a channel for the child through which he or she can raise questions and voice opinions [3,4]. The results of these actions are improved quality of life, better adherence to treatment and a decrease in depression among patients [1,3,4]. Aside from the wide developmental range of pediatric oncology patients, several barriers render communication challenging. Some of these are represented by cultural differences, complex terminology, having a different perception of which topics are more important to discuss and one’s readiness to receive bad news or misconceptions about the disease [5].

In order to help healthcare professionals to understand the importance of patient- and family-centered communication, in 2007, the National Cancer Institute described a framework for this type of communication by identifying its six main functions. These are as follows: (1) fostering the patient–clinician relationship, (2) exchanging information, (3) responding to emotions, (4) managing uncertainty, (5) making decisions and (6) enabling patient self-management [6].

Prior to the COVID-19 pandemic, some studies noted the lack of involvement or underrepresentation of the pediatric patient themselves in the communication process in regard to outcomes of the development of a neoplastic condition, while parents seemed to take the primary role. This is also reflected in decision making [7,8,9]. Themes such as wanting more information, an honest yet warm manner of communication, the management of uncertainty and emotions or maintaining hope have been among the most common [6,7,8,9,10].

This systematic review aims to analyze the latest literature on communicating the prognosis in pediatric oncology and to identify the involvement of patients, their caregivers and healthcare workers in the discussion of the prognosis, along with how the discussion might affect them emotionally. A narrow research frame was chosen for this endeavor, in order to capture the current trends in the field of communication. Despite the fact that, for the last thirty years, the main recommendation has been to involve the pediatric patient in the discussion, most studies on this topic still focus on the parents/caregivers rather than the patients themselves [3]. Hence, this review also aimed to determine whether there has been any shift in recent years regarding who the research focuses on, especially since the pandemic and its restrictions have forced the medical field to change the ways in which physicians communicate with patients and their family members.

## 2. Materials and Methods

### 2.1. Study Design

The current systematic review was included in the PROSPERO registry for systematic review protocols and followed the Preferred Reporting Items for Systematic Reviews and Meta-Analyses (PRISMA) guidelines to provide a comprehensive overview of communication aspects in pediatric oncology.

An advanced search was performed on PubMed, Scopus and the Cochrane Library using the keywords: ((prognosis communication) AND ([pediatric] OR [children]) AND ([cancer] OR [malignancy] OR [oncology])). We reviewed data from the literature, presented as original articles covering the period from 1 January 2017 to 31 October 2022 (5 years), which resulted in 374 eligible articles. After reading the abstracts, 300 studies were excluded as they did not focus on the topic. Twenty duplicates were removed using EndNote. Only original articles in English were included after further reading of the remaining studies, resulting in another 23 papers being excluded. Twenty-one final studies remained.

### 2.2. Selection Criteria

The following inclusion criteria were implemented against papers that were found through the searches: (1) full-text original work published in a peer-reviewed journal; (2) articles featuring oncologic pediatric patients and/or parents of oncologic pediatric patients and/or physicians providing care to pediatric oncology patients; (3) articles written in English. Exclusion criteria were (1) reviews; (2) commentaries; (3) editorials; (4) letters to the editor; (5) meta-analyses.

### 2.3. Data Extraction

Each title and abstract was independently reviewed by two researchers (A.B. and E.M.) in line with our inclusion and exclusion criteria. Any discrepancy between the two researchers during the screening process was settled through discussion or by the involvement of a third senior researcher (E.B.). The article was added to the entire read set if there was still any uncertainty.

The following data items were collected from the articles, regarding pediatric patients: age, sex, race, diagnosis, physician-rated prognosis, presence of the child in the initial discussion, recurrence, quality of life, death, emotional responses and treatment options.

Extracted parent characteristics were age, sex, education, marital status, parent-rated prognosis, trust, decision roles, emotional responses and opinions on communication with the healthcare provider.

Extracted physician characteristics were sex, experience, methods of communication and agreement with caretakers.

All data were extracted from article texts, tables, figures and online Appendix A, and an Excel (Microsoft, Redmond, DC, USA) database was created. Moreover, studies were split into studies with quantifiable data (QD) and studies that only mentioned a certain characteristic without providing quantifiable data (NQD)

### 2.4. Quality Assessment

Two investigators (D.D.V. and I.C.B.) independently assessed data from papers and documented findings by using the Study Quality Assessment Tools published by the NHLBI. The tools are specific to study designs and test for potential flaws in study methods or implementation. The Quality Assessment Tool for Observational Cohort and Cross-Sectional Studies was used, accessible at the following link: https://www.nhlbi.nih.gov/health-topics/study-quality-assessment-tools (accessed on 12 December 2022). Answers of “Yes” for each of the 14 questions in the tool were worth 1 point, while responses of “No” or “Other” were worth 0 points. The final quality score was then calculated. As a result, studies with a rating of 0 to 4 were deemed to be of low quality, studies with a grade of 5 to 9 were deemed to be of fair quality and studies with a grade of 10 and above were deemed to be of high quality.

## 3. Results

### 3.1. Overview

This study analyzed the results of 21 different studies, most of which were prospective (20). Eight studies were interview cohorts, 12 were survey/questionnaire-based and 1 was a mix of both interviews and surveys. The flow diagram is presented in Figure 1, while data regarding the 21 studies included are presented in Table 1.

For the studied period, 374 total records were found when searching on PubMed, Scopus and the Cochrane Library. Twenty of these records were observed to be duplicates across the studied databases, which resulted in 354 studies, whose titles and abstracts were screened as previously described in Section 2.3. This led to 200 total records being dismissed. Most (n = 79) were excluded as they did not study pediatric patients, while the rest were excluded as they did not focus on communication. This resulted in 54 records being fully analyzed. Out of these, 33 were excluded due to being letters to the editor, reviews, systematic reviews and meta-analyses, resulting in 21 final papers.

Of the total 21 studies, most (n = 17, 80.95%) were considered of fair quality, four (19.05% were of good quality and there were no poor-quality studies. The median quality score across all studies was eight, which is considered fair. There were 14 (66.67%) studies focusing only on one of the three (pediatric patients, caretakers or physicians), five (23.81%) that focused on two of the three and two (9.52%) that focused on all three groups. Parents were the most studied group, appearing in 17 (80.95%) studies, followed by children (n = 7, 33.33%) and clinicians (n = 5, 23.81%), with one (4.76%) study containing data regarding the general population.

### 3.2. Parents

Demographic data of caretakers are detailed in the Appendix A. A total of 3198 parents had quantifiable data across 17 studies. As such, parents seemed to be the most studied population. Data were not homogenous. As a result, the number of studies that contained missing or incomplete data is presented, alongside the number of studies that did not report certain categories at all (NR). Demographic data were taken from the following studies: 11, 13–17, 19–21, 24–27, 29–31 for age; 11, 13–21, 23–31 for sex; 11, 13–17, 19–21, 25–31 for education; 11, 13–22, 24–27, 29, 31 for marital status; and 11–17, 19–21, 25–29, 31 for race. The median age was 42.5, ranging from 21 to 65, as provided by studies 15, 24 and 30. Most caregivers of the studied children with cancer were aged 30–39 years (n = 1057, 39.08%); were female (n = 2486, 79.68%); had graduated from a higher form of education, such as college, professional school or university (n = 1847, 65.38%); were married or lived as a couple (n = 2479, 84.27%); and reported their race as white (n = 2250, 78.18%). Most parents were aware of their child’s prognosis (n = 2047, 92.17%). Most studies stated that primary information sources were communicated by the hospital staff, namely oncologists, nurses and psychosocial specialists (n = 15, 88.24%), followed by further reading (n = 3, 17.65%), socializing (n = 2, 11.76%) or other (n = 1, 5.88%).

Regarding trust, from the six (35.29%) QD studies [11,16,20,24,27,30], it was observed that most parents trusted their physicians/hospital/treatment highly or even completely (n = 1139, 73.01%). However, about a fifth of caretakers (n = 326, 20.90%) declared low levels of trust. Regarding parental distress, from the four (23.53%) QD studies [11,12,20,24], it was observed that most parents (n = 365, 46.68%) claimed moderate levels of distress or a need for psychosocial support. However, about a third (n = 269, 34.40%) signaled high levels. In regard to roles in decision making, the active, parent-led approach was observed the most (n = 397, 48.00%), followed by a shared variant (n =235, 28.42%), based on four QD studies [11,20,25,28]. All main characteristics related to prognosis disclosure are detailed in Table 2.

Regarding communication, in one study [24], the majority (n = 197, 90.78%) considered that the disease prognosis communication was conducted in a respectful and softened way. Less than half (n = 167, 43.15%) of the caregivers claimed to have an accurate understanding of the information given, while 15.38% directly acknowledged some form of communication barrier between themselves and the physician. In two studies [28,30], which accounted for 81 participants (65.85%), parents persisted in their intention to cure their child despite an unfavorable prognosis from the physician. Only around half of the parents reported high-quality levels of communication (n = 830, 52.73%) [11,13,16,17,20] and information (n = 892, 56.71%) [11,13,16,17,20]. Diverse racial discrepancies were observed in two studies [11,25]. This is detailed in Table 3.

The main study findings in regard to emotional status can be seen in Table 4. Data regarding the number of parents were extracted from QD studies. It can be observed that parents held a strong desire for more information (about disease and prognosis: n = 755, 86.98%; about treatment and likelihood of cure: n = 313, 78.84%). Decisional regret was observed in 105 (37.18%) patients across the same studies. The following positive emotions were observed: optimism (n = 558, 48.52%) [11,12,14,20], acceptance (n = 742, 56.30%) [11,14,20], hope (n = 227, 52.67%) [11,14,16,20]. The following negative emotions were observed: pessimism (n = 472, 36.36%) [11,14,16,20,23], depression (n = 209, 29.52%) [16,20], anxiety (n = 364, 51.41%) [16,20]. Strong spiritual beliefs were observed in two NQD studies [14,30], from which it was observed that around a third of parents (n = 103, 37.18%) were highly religious

### 3.3. Pediatric Oncology Patients

Demographic data were taken from the following studies: 11–31 for age; 11, 14–31 for sex; and 11–15, 17, 20, 21, 23, 25, 27–29 for race. These data are recorded in detail in the Appendix A. A total of 19 studies featured some form of data. However, of these 19, only seven (36.84%) focused on detailing communication characteristics in pediatric patients [13,15,19,23,24,30,31]. Again, the data were not homogenous. As a result, the number of studies that contained missing or incomplete data is presented, alongside the number of studies that did not report certain categories at all (NR). The median age was 10.27, with the range of 0–18, as provided by the following studies: 12, 13, 18, 19, 22, 24, 26, 28, 30, 31.

Pooled together, there were 3122 total children. Ten (52.63%) studies [11,13,15,16,19,23,24,28,30,31] reported whether children were present alongside their parents during the discussion regarding disease prognosis. Only about a third (n = 839, 36.02%) were. Most patients were adolescents between 13 and 18 years old (n = 839, 31.93%) and of white descent (n = 1920, 77.76%). The sex distribution was similar (M = 1378, 50.77%; F = 1336, 49.23%).

Regarding the oncologic diagnosis, data are presented in Table 5. A total of 3080 (98.65%) children had data regarding diagnosis, with only one (5.26%) study [25] reporting incomplete data. All data are presented as % from the category total. As such, most patients suffered from hematologic malignancies (n = 1434, 46.56%), followed by solid tumors (n = 1217, 39.51%). Regarding hematologic malignancies, which were present in 14 (73.68%) studies [11,13,14,15,16,17,19,20,24,25,26,27,30,31], leukemia was more prevalent (28.92%). Most solid tumors were neuroblastomas (9.88%), while the most frequent brain tumor was glioma (6.19%).

Information about quality of life, recurrences, relapses or refractory cases and mortality is provided in Appendix A. Quality of life had two types of assessment: one from “poor” to “excellent” and one from “low risk” to “high risk”. As such, low risk (≤10% chance of future limitations) was correlated with better quality of life (excellent or very good), moderate risk (10–49% chance of future limitations) was correlated with good quality of life and high risk (≥50% chance of future limitations) was correlated with poor quality of life.

The concern for quality of life was expressed only in three (15.79%) studies [15,19,20], and, for most patients, it was considered excellent or low-risk (n = 211, 48.06%). Only five (26.32%) studies [12,23,24,28,30] presented data on recurrences, relapses or refractory cases and only eight (42.11%) declared patients who were deceased [12,22,23,24,26,28,30,31]. The pooled data were compared to the grand total of children (n = 3122), resulting in 250 (8.01%) cases of recurrence, relapse or refractory tumors and 232 (7.43%) cases of patients passing.

The main study findings regarding children’s points of view in terms of communication aspects can be seen in Table 6. Data regarding parents were extracted from studies that focused especially on children (n = 7). One (14.29%) study [13] found discrepancies in children’s presence during meetings in regard to age: “ages 3–6 and 7–12 less likely to be present than infants or adolescents”. Two (28.57%) QD studies [22,28], which mentioned a total of 99 (42.49%) children and one (14.29%) NQD [19], declared that their means of prognosis disclosure toward children was more direct, while only one (14.29%) NQD [23] declared that this was done in a softened manner. Three (42.86%) NQD studies [19,30,31] presented children who had a strong desire to know more about both their disease and treatment. A role in decision making was numerically expressed in one (14.29%) study [30] featuring two (33.33%) adolescents and descriptively in another [19]. One (14.29%) QD [30] described all its children as highly religious, while another two (28.57%) NQD [14,31] only mentioned such patients.

Children expressed emotions such as pessimism (1 QD [15] and 1 NQD [19]), optimism (1 QD [15]) or high amounts of distress (1 QD [31] and 2 NQD [19,22]). In one (14.29%) study [15], children also expressed their opinions on their prognosis, out of which around half (n = 45, 45%) matched their physician’s. The management of emotions and that of uncertainty were described in one (14.29%) QD each [31], respectively [23] and in another two (28.57%) NQD [19,30]. This is detailed in Table 7.

In regard to treatment, two (28.57%) studies [28,30] featured 81 (65.85%) children who underwent an intense or even experimental treatment orientation, and 42 (34.15%) who received a low-intensity treatment or none at all. Stem cell transplantation was observed in one (14.29%) QD study [24]. Adverse reactions due to treatment were mentioned in four (57.14%) studies [19,22,28,30].

### 3.4. Physicians

A total of 11 studies had some form of data in regard to physicians, but only five (45.45%) considered them as the main focus [11,21,23,29,30]. As such, this was the least researched population. Data were not homogenous. As a result, the number of studies that contained missing or incomplete data is presented, alongside the number of studies that did not report certain categories at all (NR). Demographic data for physicians are recorded in the Appendix A. Here, we observed a total of 1558 physicians across 11 studies, with sex tending slightly toward males (n = 744, 54.23%). One (9.09%) study [11] presented incomplete data, while five (45.45%) did not report any at all [12,13,14,20,25]. In regard to the physicians’ experience or role, most were specialists or fellow doctors, who had usually amassed less than 20 years of experience (n = 307, 55.02%). One (9.09%) study [11] presented incomplete data, while five (45.45%) did not report any at all [13,17,20,21,25].

From the 11 studies featuring data from the clinicians’ side of prognosis communication, five (45.45%) [12,13,14,20,25] did not report how the communication was facilitated, whether descriptively (only word), numerical (only statistics) or mixed. Two (18.18%) studies [21,29] did not evaluate this aspect at all and hence were labeled as not applicable. From the four (36.36%) studies that had data [11,15,23,30], the most used communication method was mixing both descriptive and numerical styles (n = 184, 47.30%).

Moreover, out of the five studies that focused primarily on physicians, two (40%) observed racial discrepancies due to bias in regard to how the physician perceived the parents’ level of understanding and their desire for more information [11,29].

Agreement in regard to prognosis was poor between physicians and caretakers in two (40%) studies [11,21], and only one (20%) observed a matching prognosis and good agreement between the involved parties [23]. Most (n = 297, 57.45%) parents were observed to assess their child’s chance of recovery as very high or extremely likely. However, the physicians’ prognosis was less optimistic, being very high or extremely likely in 49.52% (n = 1097) of cases. Moreover, regarding a less favorable prognosis, this was far less common among parents than physicians (7.93% vs. 27.31%). This can be observed in Figure 2.

## 4. Discussion

This review exposes the importance of communication, by highlighting how each of the three groups take part in the discussion and the emotional states that can appear. Parents claim to want to receive as much information about the disease as possible and children also desire to have information disclosed to them and to be included in conversations, but physicians cannot always accurately predict exactly how ready the patient or the patient’s family are to receive all the information about the disease.

First of all, it is important to note that most of the studied articles are surveys or interviews. Although their contribution to this field has substantial potential, it is important to keep in mind that the resulting cross-sectional and retrospective studies can be hampered by selection, recognition and recall biases, as previously observed [32]. Secondly, as noted by Sisk et al. in 2017 and 2021 [5,8], the number of studies focusing on individual levels still heavily outweighs the number of studies focusing on higher levels, such as the study of Sisk et al. in regard to clinicians’ perspectives [29].

### 4.1. Parents

With regard to communication in pediatric oncology, our review found that parents or caregivers were the most studied population. Parents, as the legal representatives of pediatric patients, were largely aware of their child’s prognosis and had received information regarding the child’s condition mostly from the hospital staff. Although knowledge of their offspring’s disease comes as a “dizzying shock” and renders them “helpless” or “powerless” [22] or incites moderate to high levels of distress, both our review [11] and the literature [4,33] show that parents still want to know as much as possible about the diagnosis and prognosis of the disease. Systematic reviews from Hrdlickova et al., Marsac et al. and Sisk et al. [1,7,8] have also observed a consistent need for complete and even ongoing information about their child’s prognosis.

However, despite their need for more information, many parents have difficulties in understanding all the information that they are provided with. This was also noted in the systematic review from 2021 of Sisk et al., where they mentioned that parents from racial minorities and/or of lower social status reported a lower understanding of medical information [5]. A similar effect was observed by Kaye et al. in 2018, reporting that lower socioeconomic status, racial minorities and non-English language preferences were linked to parental misunderstanding of medical concepts [32]. It is therefore important for physicians to think about this aspect when communicating and check whether the given information is understood correctly or revisit the prognosis conversation; an included study [17] showed that parents were more likely to be satisfied with the prognostic communication when the subject was raised again a few months after the initial conversation. This would be a opportunity to increase the communication quality. In addition, physicians should be aware that parents who report high-quality communication and trust in their physician tend to place greater importance on implicit rather than explicit information [14].

Most parents, when asked about the single most important goal in terms of the patient’s care, chose the cure, and only a few chose quality of life [28]. Our review found some discrepancies between the prognosis estimated by the physician and that reported by the parents. Less than half of the patients’ parents had described a prognosis matching the physician’s. This emphasizes the need to consider how to deliver prognostic information regardless of the prognosis.

To ensure high quality of communication, the way in which the information is provided is of paramount importance. Parents wish for the disclosure to start gradually and for it to take place at a pace that they are emotionally prepared for [34]. They also expect the facts to be laid out in a calm, sensitive, honest and empathetic manner [1,7,8,12]. When high-quality communication is achieved, there is an increase in the trust of the medical team and also an increase in peace of mind [14]. This relationship was also noted by Sisk et al. in 2017 [8]. In our study, high levels were observed as well.

An older study by Mack et al. observed, in a multivariate analysis, that a clear explanation of what to expect during the end-of-life period, sensitive communication, speaking directly to the child when appropriate and preparing the parent for the circumstances surrounding their child’s death were all factors that were associated with higher parent ratings of physician care. Despite this, there was no correlation between parent and doctor care evaluations [35].

Although they themselves wish to receive very detailed information, when it comes to disclosing said information to their children, parents or caregivers become more apprehensive and often wish to be the first ones to receive the news, as they do not want to show any signs of weakness in front of the child, or they want to filter the amount or limit the depth of the information that the pediatric patient will receive [1,29,30]. Parents reported three factors that contributed to limited communication with their child: “information overload and emotional turmoil, lack of knowledge and skills for disclosing the diagnosis, and assumptions about burdening the child when discussing cancer” [3]. Marsac et al. describe this effect in their review from 2018 as well, as they state that many parents considered their children to be protected by avoiding discussions about unfavorable prognoses and the possibility of death [7]. Additionally, a study from 2017 by Sisk et al. found that parents of patients with a more favorable outcome wanted more details than parents whose children had less positive prognoses [36]. The role of the parent in decision making has maintained a primary position, similarly to what Lin et al. and Hrdlickova et al. reported [1,9].

In regard to expressed emotions, hope and acceptance were among the most important, similar to previously published systematic reviews by Sisk et al. and Kaye et al. [8,32]. Moreover, similarly to Sisk et al., a few studies (three in this review) addressed the topic of help in managing uncertainty [8].

### 4.2. Pediatric Oncology Patients

Despite their young age, pediatric patients wish to know their diagnosis and prognosis; they wish to understand the situation and what is likely to happen to them [14,19,31]. Delaying the disclosure of information leads to more suffering in situations where children wish to be included in the conversation [4,33,34,35]. The issue of patients wanting more information was also tackled by other systematic reviews, starting in 2017 [1,7,8,9]. This need is still persistent to this day, despite guideline recommendations.

Communicating effectively, with honesty, sensitivity and empathy, with pediatric patients has been shown to decrease their anxiety and depression, and, in the longer term, it empowers the patients and their families as well [2]. Children and adolescents also experience certain social concerns, such as the influence of treatment on their day-to-day life or their appearance, feelings of loneliness and a desire for normalcy (or for life to eventually return to how it was before their illness) [31]. These results highlight the importance that adolescent patients place on living a normal life, which is typically overlooked by clinicians as they concentrate on healing the disease [19].

It was observed that parents, in order to maintain their child’s positive attitude and hope, decided to limit the amount of information that their child received about their prognosis [1,30]. From this perspective, one of the included studies [24] showed that while almost 90% of the children received information about their diagnosis, less than half of them also received information about the disease’s prognosis. A possible explanation for this phenomenon could be that certain parents do not deem it appropriate to talk about death with their sick children, as observed in the studies of Stein et al., Marsac et al. and Sisk et al. [2,7,8].

It appears that these methods of shielding children are not as effective as the adults intend them to be. Pediatric oncology patients experience feelings of uncertainty, loneliness and threat regardless of whether they know about their diagnosis and prognosis [29], and pediatric participants in a study [33] emphasized the significance of communication and information about their health and treatment acceptance. The same types of feelings and themes were expressed in the study of Lin et al., with the authors concluding that parent-centered communication can prove disempowering and child agency should always be promoted [9].

Physicians and family members should be aware of the fact that young individuals with cancer seek additional information from other sources, such as the Internet, and patient-focused websites are not necessarily designed for adolescents and can be of inferior quality. Hence, they should encourage an open dialogue and perhaps offer guidance regarding places to acquire accurate information [19]. This was also noted by Sisk et al. in 2017, with respect to adolescent self-management, with the search for additional information being among the most important needs for self-management in this age group [8].

A study included in our analysis revealed that a quarter to a third of the included adolescents did not want information about their disease or likelihood of death [2], and given that the physician’s accuracy in predicting parental communication preferences is about 50% [25], one can only assume that the same applies to children as well. However, in another study, a number of adolescents and young adults wanted additional information regarding their health [37]. It is therefore crucial to first assess which type of information the patient is willing to receive before delivering any news. Another important aspect to consider is who will deliver the news to the pediatric patient, as some parents prefer to disclose the information themselves [1]. An honest dialogue between parents and their children with advanced cancer predicts lower child distress scores [9]. Moreover, a study from 2015 by Weaver et al. observed that parents who had a conversation with their children about the possibility of death did not regret having this conversation and emphasized an honest dialogue and inclusion in decision making, especially for adolescents [38]. Nevertheless, the number of pediatric patients involved in decision making is modest, despite guideline recommendations. The earliest studies to notice this trend were those of Sisk et al. and Kaye et al. [8,32], while, more recently, it was observed by Hrdlickova et al. and Lin et al. [1,9].

Emotionally, high distress, which can be described as feeling powerless, helpless, fearful or outright anxious, was observed in three studies. The need for emotional and uncertainty management was expressed similarly. These emotions and themes were also described in the studies of Sisk et al. and Lin et al. [8,9].

By keeping the pediatric patient at the center of our interactions, through under-standing the developmental characteristics that distinguish pediatric patients, particularly their communication styles, we can comprehend and respect their information needs and preferred levels of engagement, thereby providing truly patient- and family-centered care [9,19]. Lastly, coping is also of paramount importance, with studies observing lower dysfunctional coping mechanism rates in pediatric cancer survivors when compared to their non-oncologic peers [39,40,41,42].

### 4.3. Physicians

Communicating the diagnosis and prognosis of a malignant disease can be an extremely challenging task for healthcare providers. It is often perceived as an emotional burden that causes anxiety, especially when it comes to the patients’ or families’ reactions to the disclosed information [2,4,32,43]. An example can be found in the work of Marsac et al. from 2018, where it was observed that medical teams often refrained from discussing end-of-life care with pediatric patients until death was imminent, for a variety of reasons [7]. The study from 2017 by Sisk et al. observed that the physicians’ role was largely to provide medical information, rather than discuss emotions. Moreover, more recently, in the review from 2021 by Sisk et al., it was stated that some studies observed a tension in regard to clinicians between showing empathy and creating an appropriate emotional distance [5].

Moreover, there is a need for physicians to facilitate realistic decision making while sustaining hope by evaluating patients’ and families’ views on prognoses (which are typically unduly optimistic). It can be observed that this problem has been identified by studies before. The study of Hrdlickova observed that a physician’s understanding of a patient’s and their parents’ perspective plays a crucial role in determining an appropriate method of delivering serious news, with a preference for individualized approaches [1]. However, a recent study by Porter et al. found that they were able to implement thoughtful and effective strategies to prepare families for possible future disease progression [44]. Communication and interpersonal skills are important competencies in patient-centered care and are linked to enhanced patient health outcomes, greater patient adherence, fewer malpractice claims and higher patient satisfaction [17,19,45].

More often than not, healthcare providers would prefer their pediatric patients to receive information on their disease, depending on their comprehension level [21,30], since it has been demonstrated both in this review and in the medical literature that patients prefer to know about their condition [4,15,19,31,33,43]. This was also covered by Hrdlickova et al., Marsac et al. and Sisk et al. in their respective systematic reviews [1,7,8], while the earliest article expressing the critical role of empowering the patient by providing appropriate medical information was the one by Mu et al. [46]. In the pediatric field, this task can prove to be quite demanding, as physicians are confronted with a feeling of “split loyalties” between their patient and the patient’s family, when the family chooses to withhold information from the child [1,5]. Clinicians should be taught to recognize a child’s indications and to engage in such dialogues if they arise, but not to prompt or coerce them. Alternatively, physicians should acknowledge that each patient and parent has distinct requirements, which may manifest differently over time and in different contexts [3,44].

It is worth mentioning that, along with the aforementioned circumstances that make communicating bad news a challenging task, the literature also describes a lack of adequate training in this regard. Physicians report deficits in their formal training and skills, which lead them to feel inexperienced and uncertain about how to conduct such conversations [2,4,7,8,33]. Clinician communication training should also include techniques to elicit parents’ perspectives with sensitivity, as direct questions may not be the most effective method to establish their needs or priorities [17]. Alongside the limited time of physicians, Dylan et al. classified the lack of training in communication as a healthcare system barrier [43].

It was revealed that the parents’ need was not for more information, but rather for guidance on how to apply the information provided in the face of such unpredictability. How to help parents to cope with uncertainty and render prognostic information comprehensible requires additional study [17]. This fact was also acknowledged by the systematic review of Kaye et al., which observed that physicians who underwent a communication intervention and corresponding booster sessions demonstrated improvements in their skills during informed consent conversations [32].

Proper care for pediatric oncology patients and their families exceeds the capability of a single practitioner in the modern healthcare system. Pediatric palliative oncology care should be delivered by an interdisciplinary team to effectively meet the physical, emotion-al and spiritual needs of patients and their families. This team may include physicians, nurses, advanced practice clinicians, social workers, chaplains, child life experts and other psychosocial support personnel, who can build a comprehensive care plan that addresses the patient’s and their family’s expressed and perceived needs [4]. As observed by Mu et al. in 2015, the preparation of the family should be evaluated in light of the psychological strain brought about by the prospective loss of their healthy child. To support normal family life, health providers should improve family coping mechanisms. This may be achieved by empowering those with good attitudes toward caring for the child and assisting the family in developing the essential health-related communication skills to make the condition of the child clear [46].

### 4.4. Limitations

There are a few possible limitations to our study that should be acknowledged. One drawback is that the sample size of the chosen papers was fairly limited. Another drawback is that the data from the chosen research were varied, since some of them concentrated on different facets of prognostic communication and different methodological approaches.

Another potential limitation could be the 5-year period, as the research followed the most recent studies only. However, many older articles may remain relevant today. As the pandemic due to COVID-19 overlapped with many of the studied articles, it is important to note that it could have had an impact on the number of studies produced during this period.

Another important limitation, which was observed in other studies as well, is the scarcity of data on the siblings of pediatric oncologic patients and limited data on other members of the clinical team, such as nurses, psychologists, etc.

Two researchers were assigned to evaluate the quality of the chosen studies, therefore lowering the risk of potential bias, such as selection, missing data or measurement bias. This was done to combat the inherent biases of these types of articles. The authors advocate for meta-analysis research on the subject to further lower the risk of bias and improve the data accuracy.

## 5. Conclusions

In the past few years, there has been ongoing interest in the subject of communication in all fields, including medicine. Our study’s purpose was to examine the most recent literature on communicating prognosis in pediatric oncology, pinpoint the challenges encountered and highlight the areas where further research is required.

To improve the quality of communication, healthcare workers should receive professional training. This would not only help them to communicate with their patients and their families, but it would also help them to manage the emotional toll of having to deliver bad news. At the beginning of each conversation, the physician should assess the needs of the patient and their family regarding the amount of information that they are ready to receive on the subject.

Although children may or may not understand the severity of the situation, they seem to want to know more and be included in conversations about their conditions. Many can suffer when they feel that they are not properly informed. Moreover, this subject ought to be studied extensively, and pediatric patients should be offered the opportunity to voice their opinions, as they deserve to have a voice and have their direct perspectives included in studies, instead of having their thoughts filtered through the mind of an adult, whether a family member or a healthcare worker.

## Figures and Tables

**Figure 1 children-10-00972-f001:**
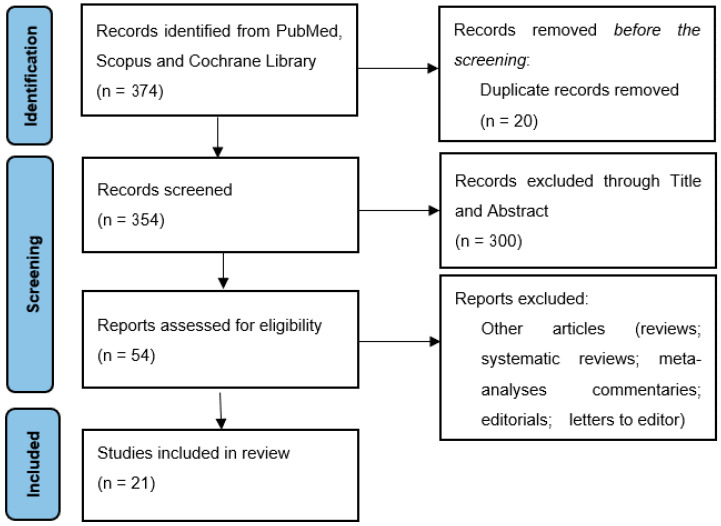
PRISMA flowchart for the selection process.

**Figure 2 children-10-00972-f002:**
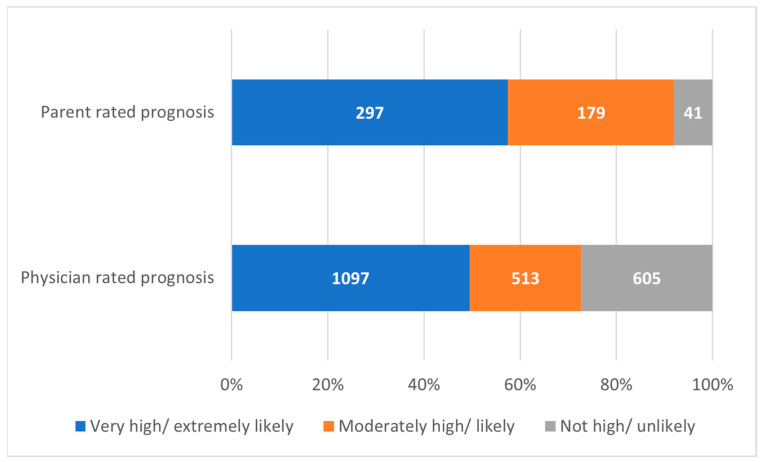
Parent- and physician-rated prognosis for pediatric oncologic patients.

**Table 1 children-10-00972-t001:** Studies included in the analysis.

Nr.	Study Type	Country	Year	Focus on	Quality Score
1 [11]	Prospective questionnaire cohort	USA	2017	Parents and clinicians	good
2 [12]	Prospective questionnaire/interview cohort	USA	2019	Parents	fair
3 [13]	Prospective questionnaire cohort	USA	2019	Parents and children	fair
4 [14]	Prospective questionnaire cohort	USA	2017	Parents	fair
5 [15]	Prospective questionnaire cohort	USA	2021	Parents and children	fair
6 [16]	Prospective questionnaire cohort	USA	2018	Parents	fair
7 [17]	Prospective questionnaire cohort	USA	2017	Parents	fair
8 [18]	Prospective interview cohort	UK	2020	Parents	good
9 [19]	Prospective interview cohort	USA	2017	Children	fair
10 [20]	Prospective questionnaire cohort	USA	2018	Parents	fair
11 [21]	Prospective questionnaire cohort	Republic of Korea	2018	Clinicians and general population	good
12 [22]	Retrospective interview cohort	UK	2017	Parents	fair
13 [23]	Prospective interview cohort	USA	2022	Parents, children and clinicians	fair
14 [24]	Prospective questionnaire cohort—population-based nationwide survey	Sweden	2022	Parents and children	fair
15 [25]	Prospective questionnaire cohort	USA	2020	Parents	fair
16 [26]	Prospective interview cohort	USA	2020	Parents	fair
17 [27]	Prospective questionnaire cohort	USA	2020	Parents	good
18 [28]	Prospective questionnaire cohort	USA	2020	Parents	fair
19 [29]	Prospective interview cohort—focus groups	USA	2021	Clinicians	fair
20 [30]	Prospective interview cohort	Mexico	2017	Parents, children and clinicians	fair
21 [31]	Prospective interview cohort	UK	2018	Children	fair

**Table 2 children-10-00972-t002:** Main characteristics related to prognosis disclosure in parents.

Prognosis Disclosure		Trust		Parental Distress		Decision Making	
Total	2221	Total	1560	Total	782	Total	827
Yes	92.17%	High levels	73.01%	High	34.40%	Parent-led	48.00%
No	7.83%	Moderate levels	6.09%	Moderate	46.68%	Oncologist-led	23.58%
		Low levels	20.90%	Low/None	18.93%	Shared	28.42%
NR *	5	NR *	9	NR *	10	NR *	8
QD *	12	QD *	6	QD *	4	QD *	4
NQD *	0	NQD *	2	NQD *	3	NQD *	5

*: % of 17 total studies with parental data; NR: not reported; QD: studies with quantifiable data; NQD: studies only mentioning said characteristics, without providing quantifiable data.

**Table 3 children-10-00972-t003:** Main parental study findings in regard to prognosis communication.

	Parents (%)	QD	NQD
Accurate understanding	43.15%	3	3
Acknowledgment of barriers	15.38%	1	1
Desire for more information about disease	86.98%	3	-
Desire for more information about curability	78.84%	2	-
Parent rated high-quality information	56.71%	5	-
Parent rated low-quality information	40.24%	5	-
Parent rated high-quality communication	52.73%	5	-
Parent rated low-quality communication	46.32%	5	-
Parent rated the usefulness of study	62.07%	1	-
Respectful/sensitive/softened manner of communication	90.78%	1	3
Racial discrepancies in communication	-	-	2

QD: studies with quantifiable data; NQD: studies with unquantifiable data, studies that do not provide quantifiable data.

**Table 4 children-10-00972-t004:** Main study findings in regard to emotional status expressed by caregivers.

	Parents (%)	QD	NQD
Pessimism or struggle	36.36%	5	-
Optimism	48.52%	4	1
Depression	29.52%	2	-
Anxiety	51.41%	2	-
Hope	52.67%	2	2
Acceptance or peace of mind	56.30%	4	1
Help in managing uncertainty	66.10%	2	1
Decisional regret	23.33%	2	-
Strong religious/spiritual beliefs	37.18%	2	2

QD: studies with quantifiable data; NQD: studies with unquantifiable data, studies that do not provide quantifiable data.

**Table 5 children-10-00972-t005:** Children’s oncologic diagnosis.

Total * n = 3080	%	QD		%	QD		%	QD
**Hematologic** **Malignancies**	**46.56%**	**14**	**Solid Tumors**	**39.51%**	**14**	**Brain** **Tumors**	**13.93%**	**13**
*Leukemia*	28.92%	4	Ewing sarcoma	8.64%	2	Medulloblastoma	5.42%	2
AML	16.07%	3	Rhabdomyosarcoma	6.17%	2	Glioma	6.19%	2
ALL	12.85%	3	Neuroblastoma	9.88%	3	Teratoma	2.32%	1
*Lymphoma*	17.64%	2	Melanoma	2.47%	1			
Hodgkin	8.82%	1	Wilms tumor	1.23%	1			
Non- Hodgkin	8.82%	1	Germ cell tumor	2.47%	2			
			Hepatoblastoma	1.23%	1			
			Colon adeno-carcinoma	2.47%	1			
			Osteosarcoma	4.94%	1			

QD: studies with quantifiable data; AML: acute myeloid leukemia; ALL: acute lymphocytic leukemia; *: % of 19 total studies with pediatric data.

**Table 6 children-10-00972-t006:** Data on prognosis communication regarding children.

	Children (%)	QD	NQD
Discrepancies in meeting presence due to age	-	-	1
Direct disclosure	42.49%	2	1
Softening the message	-	-	1
Desire to know more about disease	-	-	3
Desire to know more about treatment	-	-	3
Involved in decision making	33.33%	1	1

QD: studies with quantifiable data; NQD: studies with unquantifiable data, studies that do not provide quantifiable data.

**Table 7 children-10-00972-t007:** Main study findings in regard to emotional status expressed by children.

	Children (n, %)	QD	NQD
Pessimism	15.00%	1	1
Optimism	43.00%	1	-
Distress and/or anxiety	66.67%	1	2
Required emotional management and reassurance	100.00%	1	2
Need for management of uncertainty	47.50%	1	2
Strong religious/spiritual beliefs	100.00%	1	2

QD: studies with quantifiable data; NQD: studies with unquantifiable data, studies that do not provide quantifiable data.

## Data Availability

Data available on request.

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
