# Peer review of "Prognosis Communication in Pediatric Oncology: A Systematic Review"

_children, 2023, doi:10.3390/children10060972_

Round 1
Reviewer 1 Report
Overview:
In this manuscript “Prognosis communication in pediatric oncology, a continuous challenge – a systematic review,” the authors aimed to gather and synthesize recent literature on prognostic communication in pediatric oncology and identify communication challenges experienced by patients, caregivers, and clinicians. Please see below comments with recommendations for the authors to consider with the goal of strengthening this manuscript.
1. Recommend streamlining the title: “Prognosis communication in pediatric oncology: a systematic review.”
2. In the introduction, the authors state that “open communication” is the preferred approach; I caution the authors to be careful about making blanket assumptions, as preferences for prognostic communication vary significantly across different sociocultural contexts.
3. In both the introduction and the conclusion, authors state that the aim of this review was to synthesize existing literature on communication of diagnosis and prognosis; however, this broadly scoped aim is not reflected in the search methods, which appear to query literature specific to prognostic communication and not communication at diagnosis (which encompasses a different subset of communication).
4. Did a medical librarian assist with the design and execution of the advance Boolean search strategy? I worry that use of the phrase “prognosis communication” as a specific and discrete search concept may have missed a number of articles, as this phrase is neither standardly nor consistently used in the literature. Typically, “prognosis” and “communication” would be entered as separate concepts within the advanced search strategy, using Boolean logic (“AND”) to narrow the search to articles that include the intersection of these concepts. The authors need to justify their approach here, as well as provide evidence that they sufficiently evaluated and addressed potential missingness in the search strategy.
5. The results section is very comprehensive, and it may also be a bit overwhelming for readers (I’ve not seen a paper with >10 tables in the results section before, and I’m wondering if the paper might be more compelling if the authors were a bit more selective in which results they wish to put forward, instead of giving every piece of information equal weight).
6. Encourage the authors to make a clearer distinction between the results and discussion sections. There is a lot of “rehashing” the results in the discussion section, which can come across as redundant and distract from the main goal of the discussion – which is to focus on why the findings matter and how this review can help readers improve their future clinical practice or advance research in communication skills training. The discussion, as written, is quite long and heavily invested in repeating the findings; recommend a significant revision to emphasize the key findings that should inform specific actionable changes in practice, interventions, or future research, with specific calls to action in each of these categories.
7. Be careful not to overstate findings in the conclusion section. For example, the authors make statements about children knowing more than adults believe them to know – which may be true, but it’s not wholly supported by data presented in this specific paper (etc).
8. Encourage the authors to revise the manuscript with an eye for scientific editing. There are quite a few run-on sentences or sentences that lack the correct noun-verb structure, which make parts of the paper challenging to follow. The overall message of the paper would be strengthened by a comprehensive revision focused on shortening sentences and streamlining syntax; recommend partnership with a trained scientific editor to assist with this process.
Author Response
Thank you for your time and support. The comments and recommendations have been addressed in the attached file.

Reviewer 2 Report
The authors present a “Systematic review” of prognostic communication (which they call prognosis communication) from the last 5 years. They have identified most of the key articles, however are missing a few that I think are relevant, and also have drawn what seems to be an arbitrary line at 5 years. Communication science has not changed so rapidly that an article 6 years ago would be outdated, so I would encourage the authors to justify the narrow window of time, which mean that some important articles were not included. The results section has too many subjective statements made in it that belong in a discussion section, and it also is formatted as too many tables, with lots of long lists of numbers and percentages. The data that are trying to be highlighted are difficult to find with this formatting. Overall, I commend the authors for trying to summarize this important subject for pediatric oncology as a field, but the manuscript would be strengthened by attention to some of the comments below.
Comments & Questions:
1) Why did the authors limit this to the last 5 years? Communication science takes time, and many vital articles about it that remain relevant today are more than 5 years old. I would suggest explaining/justifying the narrow window of time, acknowledging what is missing.
2) The results section contains a lot of subjective interpretation and qualification statements, rather than objectively presenting the data and then leaving discussion to the discussion section. I would urge more attention to this during a revision.
3) The results section data are presented in a very difficult way to follow. It seems to be sliced in so many ways (based on # of studies, # of respondents within each study, percentages that appear to reflect the % of the respondents in that specific study and not the overall % of respondents from the group of studies being referred to, etc. It makes it difficult to follow and also is distracting when trying to figure out what the highlights of the results actually are. Consider reading and re-formatting.
4) There are simply too many table, and too much data presented. I would suggest narrowing this to present the data that are most relevant, not every piece of data that was compiled.
5) Discussion section: because of the overwhelming and confusing presentation of the results section, I had to go back and forth between the discussion section and results section to figure out whether the data the discussion was summarizing was even present in the results. Otherwise, the paragraphs seem more like a summary of the literature (what might be found in a background section), rather than referring to analysis that was done by the authors as part of a “systematic” review.
For some specific details/minor points:
1) Abstract Background: This is a single run-on sentence. Consider breaking this up into separate sentences.
2) Abstract methods: This is not a complete sentence, please consider revising to use a verb such as “we conducted” into the sentence, making it “Following PRISMA guidelines, we conducted an advanced search…”
3) Abstract results: Consider removing the words/sentences about the eligibility, screening, etc. Those details can remain in the manuscript. The results section should be the main results that support your conclusions. As written, the only result presented is that “most studies (17) focused on the caregivers…” By removing the prior sentence or 2, the authors could add more of the results.
4) Introduction, sentence 1: While communication is important for establishing rapport, the most important thing about it is that it is vital to effectively communicate accurate information so that patients know what is going on and can make informed decisions about their (or their surrogate’s) healthcare. Something about this should be included in the first sentence, not just focusing on establishing rapport.
5) Lines 54-56: Please add citation to this line, as it says “Recent studies” but then doesn’t cite those studies.
6) Line 63: “ardent” does not appear to fit here. Consider a simpler word/term.
7) Methods, lines 88-90: What does it mean to “analyze the latest literature”? The second part, identifying challenges encountered by various stakeholders is a more clear focus of the reading of each article. But “analyzing” is vague and it isn’t clear to me what exactly was being looked at/for with the data extraction.
8) Line 100: typo of word “ad” instead of “and”
9) Line 104: word “diagnostic” doesn’t make sense in context of this sentence.
10) Line 137-138: Suggest commenting on these other reviews/systematic reviews. In doing their own systematic review, did the authors find that similar reviews had already been done? If so, how was the approach of this review different and/or what does it add?
11) Line 143-144: It is not clear what this sentence means, with regard to “single”, “double” and “triple”. I had to re-read the methods to try to figure it out and couldn’t.
12) Lines 144-147: The formatting of the numbers and percentages should be consistent. It is confusing as written, and 4.76 doesn’t have a % next to it.
13) Line 147: The comment about the mean quality score belongs after the sentence about the quality scores that begins the paragraph.
14) Table 2: The table is confusing because numbers do not appear directly across from the row header. Please re-format, such as with different shading or lines between cells so that it is clear what is in the same row.
15) Line 158: What is defined as a “higher form of education”? That is not the standard language used in most US-english manuscripts, so consider a simple explanation (at least college/university?)
16) Lines 161-168: This paragraph is just a long list of variables and whether they “acknowledged missing data” or “did not report it at all”. It’s not clear to me what this data adds, but if needed please consider reporting it in a better way.
17) Line 176: This statement “one of the most important characteristics in prognosis communication is trust” is an opinion, and should not be in the Results section. It should be moved to the discussion.
18) Lines 180-181: Again, this paragraph starts with subjective information rather than simply reporting the objective data from their review.
19) Lines 185-188: Again, this paragraph starts with subjective information rather than simply reporting the objective data from their review.
20) Line 204: Where did the “main study outcomes or themes” come from? Who made that determination? I couldn’t find anything in the methods section about how “themes” would be identified.
21) Lines 233-238: The total number of children was 3122, but then the subsets reported (% of children alongside their parents, % of adolescents between 13-18, white descent, and sex distribution) do not have percentages or totals that add up to 3122. This becomes distracting to follow.
22) Lines 268-274: This paragraph does not make sense. It is not clear what is meant by “diseases impact on the patient”. What does it mean for quality of life to be “low risk”?
23) Lines 377-380: This refers to a “prior qualitative study” but then cites 2 different studies. Additionally, it reports 3 factors, but “information overload and emotional turmoil” are distinct entities.
24) Line 383: remove the fraction ¾ from the text. This should be words or percentage.
25) Line 411: None of the citations is a statement made by the American Academy of Pediatrics. While some are published in the journal “Pediatrics” none are AAP position statements, and the citations provided (3,15) are not.
Citation formatting: Citation #3 has the authors names as first middle last, rather than last, initials.
Additional Citations suggested:
-Sisk BA, Fasciano K, Block SD, Mack JW, “Longitudinal Prognostic Communication Needs of Adolescents and Young Adults With Cancer”, 2020, Cancer.
-Porter AS, Chow E, Woods C, Lemmon ME, Baker JN, Mack JW, Kaye EC. “Navigating prognostic communication when children with poor-prognosis cancer experience prolonged disease stability”, 2022, Pediatric Blood & Cancer.
Author Response

(The authors gave the same response as above.)

Round 2
Reviewer 1 Report
Thank you for the opportunity to review this revised manuscript. I appreciate the efforts invested by the authors to strengthen this paper. I continue to have concerns, similar to those raised in the original review, which were not wholly addressed. I’ve reiterated my main two concerns below.
1. The authors did not clarify whether a medical librarian assisted with the design and execution of the advance Boolean search strategy. I remain concerned that use of the phrase “prognosis communication” as a specific and discrete search concept likely missed a number of articles, as this phrase is not standardly nor consistently used in the literature. Typically, “prognosis” and “communication” would be entered as separate concepts within the advanced search strategy, using Boolean logic (“AND”) to narrow the search to articles that include the intersection of these concepts. For perspective, a quick search for the term “prognosis communication” yields ~50 hits, whereas a search for “prognostic communication” (a more commonly used phrase) yields ~80 hits. Conversely, a search for “prognosis” and “communication” yields a return orders of magnitude higher. The authors need to justify their Boolean approach here, as it a bit atypical, and may introduce bias in the sample.
2. There continues to be quite a bit of “rehashing” results in the discussion section. As before, I recommend revisions to emphasize just a few key findings that can inform specific actionable changes in practice, interventions, or future research, with specific calls to action in each of these categories.
Author Response
Thank you for your comments, please find the responses attached.

Reviewer 2 Report
I appreciate that the authors have revised the manuscript to address the smaller grammatical and typographic concerns and comments that I made. However I still think that there are outstanding questions about the overall approach to the study and manuscript that I think should be addressed more comprehensively to make this a more consequential addition to the communication literature.
Rationale for focus on last 5 years: I would still recommend that there be more acknowledgment of communication science prior to the study window, including, but not limited to, justifying why the authors focused on the latest literature. For example, the authors could either add a new paragraph or add information to the last paragraph of the introduction with a summary of what prior systematic reviews have found, and explain what they hoped to achieve by analyzing the most recent 5 years. Do they anticipate a change? Were there specific findings or unresolved questions from prior reviews that they hoped to see research on? Did prior ones also comment on the emphasis on parents rather than patients or physicians?
Goals of the study: Other than summarizing, it is not clear what specific objectives the authors hoped to achieve in doing this. The authors conclude the introduction saying that they wanted to “analyze the latest literature” and “identify the challenges encountered”… If that was their focus, then the results and discussion should emphasize/highlight what they found related to challenges. As it is written now, there is not a logical flow of the results and discussion that depict how they achieved that aim.
For example, another review of the literature (Hentea et al, 2018, PBC: PMID 29667725) had this focus to their review, and then the results and discussion clearly achieved this mapping of communication factors:
“In order to help pediatric providers apply these findings to clinical practice, we mapped the communication factors surrounding parent–provider communication preferences at the time of diagnosis and highlighted emerging concepts.”
Results section: I still think the results tables and paragraphs summarize lots of statistics and yet are not logically presented in a way that match the scope/aims of the authors analysis. Perhaps a clearer defined objective would guide a more logical and focused display of relevant results. Not every piece of data collected needs to be presented if not relevant to the analyses, and below include suggestions for data that the authors could consider omitting.
Tables in general: I feel these large, detailed tables (tables 4, 8, 10) would be better as supplemental tables, if included at all. The main tables within the text should really support the main conclusions and focus of the study, but not all data collected needs to be in the manuscript. The discussion session refers to so little of the results presented, that it suggests to me that the results presented need to be cut down significantly to only the pertinent positives/negatives related to the major study objectives and findings. The rest can be in supplements.
Figures in general: It seems that data was presented/changed to figures in order to reduce the number of tables. But the 3 figures represent data that I’m not sure are helpful to represent in a visual figure, or potentially in the main manuscript at all. They do not tie to the main study findings/conclusions, so could perhaps be put into supplemental data.
Discussion in general:
-This still reads like a summary of the literature that might fit in a background section, rather than a discussion of the objective findings/analysis of the systematic review (the study itself). I would refer to another systematic review of this literature (e.g. one before 2017 that the authors said they found in their preparation) for an example of how this type of discussion could appear and build upon the results section.
-Examples include:
--Hentea et al, “Parent-centered communication at time of pediatric cancer diagnosis: A systematic review”, Pediatric Blood and Cancer. 2018 Aug;65(8):e27070, PMID 29667725.
--Sisk et al, “Communication in pediatric oncology: state of the fiekd and research agenda.” Pediatric Blood and Cancer, 2017; 00: e26727. PMID 28748597
Figure 1: Parent-rated prognosis: It is unclear to me what the point of reporting this is. Other than reporting the lack of matching with the physician’s prognosis (not in the figure but in the text above), there is no way to interpret what the parents’ estimates mean since we don’t know the “truth” for these patients. So this figure does not clearly help highlight data relevant to their analysis.
Table 4: study findings. This table still doesn’t have enough explanation in the text of what each line refers to. “Main study findings observed can be seen in table 4.” is not enough context for the reader to look at the table in intuitively understand what is being presented. There is so much data presented, and yet the significance of any of it is challenging to interpret. For example, looking at the table and seeing line “Provided validation/family self-management” 140, 89.74%... How is a reader supposed to know what that means? I can guess it means that that 140 parents participated in the studies that assessed for whether communication provided validation/family self-management, and of those 89.74% reported that communication provided validation/family self-management. Or was the 140 out of another number (the denominator) to make up 89.74% of that overall number?
Table 6 and sentence referring to solid tumors:
“Most solid tumors were neuroblastomas (n= 112, 78.87%).” This doesn’t make sense. 112 is not 78.87% of the 1217 solid tumors that were reported, it would be 9.2%. When presenting this much data, it should be presented in a way that make intuitive sense to the reader without explanation. Otherwise I would suggest omitting that level of detail or coming up with a clearer way to explain it (for example with an asterisk/footnote, in the text, etc.).
Figure 2: I don’t know what the goal of reporting this information is, specific to the objectives and findings of the study.
-Additionally, the order in which the response are presented is confusing. It looks like the authors are reporting by likelihood of cure on the left, and terminal on the right, but terms like “favorable” and “unfavorable” don’t fit onto the same scale as the other 5 categories (extremely likely, very likely, moderately likely, less likely, terminal). So instead of depicting it that way, it would make more sense to go left to right based on the # of responses.
Use of “Most” for statistics that are <50% in abstract and results section: “Most parents reported… and then lists a number of things that are <50% (leading role in decision making (48%), moderate distress level (46.68%), and a few others). It seems difficult to refer to “most parents”, as <50% doesn’t typically constitute “most”. Though I think they are saying that of the responses, those had the largest #.
Discussion paragraph 1:
-“physicians cannot always accurately predict exactly how ready the patients or the patient’s family are to dive deep into the details of a disease. Once all those participating in the conversation manage to be on the same page, the reported quality of the conversation increases alongside trust and peace of mind increase as well.”
-It is not clear to me what evidence supports this claim. I do not see this reported in the study results, and the authors are not citing other evidence/papers that report this.
-The sentence is also not grammatically correct and needs to be edited.
Discussion section on pediatric patients:
-The outline of this section, and the discussion in general, does not appear related to the results presented. It reads like a summary of the authors general takeaways from the literature. I recognize that there were a limited number of studies that included the pediatric patients, but the discussion section should be a summary of the results that the authors found in doing the review, or reflect their analysis. So the discussion should either change to follow the results, or different results should be presented to justify/support the discussion.
Casual language: throughout the manuscript there are many colloquial phrases that do not add to the meaning of the manuscript and/or are grammatically incorrect. I believe these should be removed. Examples include (but not the only ones):
-Line 584: “last but not least”.
-Line 599-600: “physicians should also bear in mind that”
Grammar and formatting: there are still many grammatical errors and formatting changes that should be addressed. 2 examples (but not all) include:
-Line 347: “this can be observed visually in Figure 1.” Instead just put “(Figure 1)” before the period in the prior sentence.
-Line 408-409: “Only about a third (n= 839, 36.02%) were.” This is an incomplete sentence. It seems clear that it is referring to the prior sentence (the #/% of kids who were present during discussion), but it is grammatically incorrect.
Author Response
Thank you for your comments, the responses have been attached.

Round 3
Reviewer 2 Report
I appreciate the significant revisions to this draft, which has made for a much improved manuscript, in particular the results and discussion sections.
I still don’t think the authors have put this review in the context of prior reviews. While they responded to my prior review comment about the lack of acknowledgement of prior reviews, they did not add any text or content to the manuscript (in introduction or discussion). So, a reader of the paper (if published) will be left with the same questions that I had as a reviewer. My prior comment remains:
I would still recommend that there be more acknowledgment of communication science prior to the study window, including, but not limited to, justifying why the authors focused on the latest literature. For example, the authors could either add a new paragraph or add information to the last paragraph of the introduction with a summary of what prior systematic reviews have found, and explain what they hoped to achieve by analyzing the most recent 5 years. Do they anticipate a change? Were there specific findings or unresolved questions from prior reviews that they hoped to see research on? Did prior ones also comment on the emphasis on parents rather than patients or physicians?
Outside of this larger point that remains unaddressed (and to me is a major omission from the manuscript), there are only a few smaller comments/questions:
1. It appears that they have updated the number of records reviewed, screened, excluded. I do not see an explanation for this, did they re-review or was this an error in prior submitted versions?
2. Methods, section 2.1: The last sentence of this section was identical to the last sentence of the Introduction. The last sentence of the introduction has changed (in response to prior recommended revisions), but this sentence was kept the same. It makes sense to either remove this sentence entirely (recommendation since duplicating the same sentence isn’t needed), or update it to mirror the sentence in the introduction.
This systematic review aims to study recent literature on communicating prognosis in pediatric oncology and identifies the challenges encountered by the patients, their care- givers and the healthcare workers involved in the discussion.
3. 3.3 Children: This section still reports 1217 children with solid tumor diagnosis, and only 117 Neuroblastoma, but it says this is 31.3%. 117 of 1217 would be ~10%, not 31%. It seems that there are missing patients unaccounted for. Perhaps ~900 have solid tumors that were “not reported”?? If so, this should be included in the table and the percentages adjusted. But it doesn’t make sense to report the percentages if they are not accurate. And it doesn’t make sense to report percentages of “known” diagnoses, unless it is explicitly stated that is what the authors are doing.
4. Table 5: Has “NR = Not Reported” in the key at the bottom, but the table itself doesn’t have any columns/fields that say NR.
5. Consider adding a limitation acknowledging whether/how the COVID pandemic might have impacted study/studies of prognostic communication, since it overlapped much of the studied window of the review.
Author Response
Thank your for your continuous support. The response has been attached.
